# New SiO_2_/Caffeic Acid Hybrid Materials: Synthesis, Spectroscopic Characterization, and Bioactivity

**DOI:** 10.3390/ma13020394

**Published:** 2020-01-15

**Authors:** Michelina Catauro, Federico Barrino, Giovanni Dal Poggetto, Giuseppina Crescente, Simona Piccolella, Severina Pacifico

**Affiliations:** 1Department of Engineering, University of Campania “Luigi Vanvitelli”, Via Roma 29, I-813031 Aversa, Italy; federico.barrino@unicampania.it; 2Ecoricerche, Srl, Via Principi Normanni, 81043 Capua (CE), Italy; giogiodp@hotmail.it; 3Department of Environmental, Biological and Pharmaceutical Sciences and Technologies, University of Campania “Luigi Vanvitelli”, Via Vivaldi 43, 81100 Caserta, Italy; giuseppina.crescente@unicampania.it (G.C.); simona.piccolella@unicampania.it (S.P.); severina.pacifico@unicampania.it (S.P.)

**Keywords:** caffeic acid, sol–gel technique, FT-IR spectroscopy, radical scavenging capacity, antibacterial activity

## Abstract

The sol–gel route represents a valuable technique to obtain functional materials, in which organic and inorganic members are closely connected. Herein, four hybrid materials, containing caffeic acid entrapped in a silica matrix at 5, 10, 15, and 20 wt.%, were synthesized and characterized through Fourier-Transform Infrared (FT-IR) and Ultraviolet-Visible (UV–Vis) spectroscopy. FT-IR analysis was also performed to evaluate the ability to induce the hydroxyapatite nucleation. Despite some structural changes occurring on the phenol molecular skeleton, hybrid materials showed scavenging properties vs. 2,2-diphenyl-1-picrylhydrazyl (DPPH) radical and 2,2′-azinobis-(3-ethylbenzothiazolin-6-sulfonic acid) radical cation (ABTS•^+^), which was dependent on the tested dose and on the caffeic acid wt.%. The SiO_2_/caffeic acid materials are proposed as valuable antibacterial agents against *Escherichia coli* and *Enterococcus faecalis*.

## 1. Introduction

Caffeic acid, also known as 3,4-dihydroxycinnamic acid, is a C_6_-C_3_ phenol compound, produced by plants either from phenylalanine or tyrosine through the shikimate pathway of secondary metabolism, and it is a representative of the cinnamic acid (or phenylpropanoid) class [1,2]. It becomes a part of the human diet, included in several vegetables (e.g., potatoes, carrots, olives, coffee beans) and fruits (e.g., blueberries, apples, cider) [3,4,5]. It can be found as a monomer, in esters with small organic acids, sugars, and amides, or as a dimer or trimer and flavonoid derivative; furthermore, it can be linked to proteins and other polymers in the cell wall [6].

In addition to the antioxidant capacity, related to the catechol moiety on its molecular skeleton, a number of other biological activities, demonstrated in vitro and in some cases confirmed in in vivo experiments, are ascribed to caffeic acid, including antibacterial, anti-carcinogenic, antiviral, anti-inflammatory, immunomodulatory, anti-atherosclerosis, and cardioprotective properties [7].

Due to its healthy properties, caffeic acid was embedded in food active packaging films prepared from water-soluble chitosan (*N*,*O*-carboxymethyl chitosan) and water-soluble methylcellulose, with the aim of enhancing the antioxidant and antibacterial activity [8,9,10]. In the biomedical field, CA-g-P(3HB)-EC composites were synthesized, by grafting caffeic acid (CA) onto the poly(3-hydroxybutyrate) ethyl cellulose (P(3HB)-EC) based material, as promising biomaterials in infection-free wound dressings for burns and/or skin regeneration [11]. Caffeic acid-loaded CaP nanocomposites were developed as controlled drug release carriers, and their cytotoxicity was evaluated with respect to human osteosarcoma [12].

In the synthesis of functional materials, solid-state chemistry is combined with several solution techniques, including coprecipitation, hydrothermal processing, solvothermal methods, and sol–gel chemistry. In particular, sol–gel chemistry shows a number of advantages, including the ability to produce a solid-state material from a chemically homogeneous precursor [13]. The sol–gel route is a consolidated technique in our laboratories, and it was effectively applied with the aim of obtaining new biocompatible and bioactive organic–inorganic hybrid materials, in which different natural compounds were embedded in a silica matrix, with or without the addition of polymers (e.g., polyethylene glycol, poly(ε-caprolactone)) [14,15,16,17,18,19].

In the present work, caffeic acid represented the organic constituent of hybrid materials, whereas the silica matrix was the inorganic constituent. Four materials were synthesized, which differed in terms of the amount of caffeic acid incorporated (5, 10, 15, and 20 wt.%). Their chemical characterization was achieved through Fourier-transform infrared (FT-IR) and Ultraviolet-visible (UV–Vis) spectroscopy. The FT-IR technique was also employed to evaluate the bioactivity of the hybrid materials, after soaking them in simulated body fluid (SBF). 2,2-diphenyl-1-picrylhydrazyl (DPPH) radical and 2,2′-azinobis-(3-ethylbenzothiazolin-6-sulfonic acid) (ABTS) radical cation tests allowed us to assess in vitro preliminary antiradical properties of all samples. Furthermore, *Escherichia coli* and *Enterococcus faecalis* were used as Gram-negative and Gram-positive models to investigate the antibacterial properties.

## 2. Materials and Methods

### 2.1. Sol–Gel Process

The materials under study were obtained through a sol–gel synthetic route, in which the inorganic matrix was represented by silica gel. It was obtained by adding tetraethyl orthosilicate (TEOS, Si(OC_2_H_5_)_4_) (Sigma-Aldrich, Milan, Italy) to a solution containing nitric acid (≥65%, Sigma-Aldrich, Milan, Italy), distilled water, and ethanol 99% (Sigma-Aldrich, Milan, Italy) under stirring. The acidic environment favors the kinetics of hydrolysis and condensation reactions and, thus, affects the microstructural properties of the inorganic matrix [20]. The structural characteristics were also influenced by the H_2_O/alkoxide molar ratio. The molar ratio used for the reagents was TEOS:HNO_3_:EtOH:H_2_O = 1:1.7:6:2.

The organic counterpart of the hybrid materials was the phenolic compound caffeic acid (Sigma-Aldrich, Milan, Italy), dissolved in ethanol and then added dropwise to the synthesized silica sol under stirring. The stirrer was stopped after 20 min and, after gelation at room temperature, the products were air-dried at 40 °C for 24 h to remove the residual solvent, without compromising the thermal stability of the embedded phenol. The amount of caffeic acid varied from 5 to 20 wt.% (Figure 1).

### 2.2. FT-IR Spectroscopy

Fourier-transform infrared (FT-IR) transmittance spectra were recorded in the 400–4000 cm^−1^ region using a Prestige 21 (Shimadzu, Tokyo, Japan) system, equipped with a deuterated triglycine sulfate (DTGS) detector with potassium bromide windows, with a resolution of 4 cm^−1^ (45 scans). The FT-IR spectra were processed by Prestige software (IR solution, version 1.50, Shimadzu, Tokyo, Japan). A Specac manual hydraulic press (Orpington, UK), equipped by a cylindrical holder, was used to press sample powders (200 mg, 1 wt.% in KBr) until a disc (13 mm diameter, 2 mm thickness) was obtained. 

### 2.3. UV–Vis Spectroscopy

Firstly, 100 mg of each powdered material underwent solid–liquid ultrasound-assisted extraction (UAE) in an ultrasonic bath (Advantage Plus model ES, Darmstadt, Germany), using 2.0 mL of an hydroalcoholic solution (EtOH:H_2_O 1:1, *v*:*v*) as the extracting solvent. After 1 h, the samples were centrifuged at 4500 rpm for 5 min, and the supernatants were collected and dried under vacuum. The whole process was repeated twice. Immediately before recording UV–Vis spectra, samples were dissolved in pure ethanol to obtain a 1.0 mg/mL final concentration. UV–Vis spectra were acquired in the range 200–600 nm using a UV-1700 double beam spectrophotometer (Shimadzu, Kyoto, Japan).

### 2.4. Bioactivity Tests

The bioactivity of the hybrids materials was evaluated by an in vitro apatite forming-ability test, carried out following the procedure of Kokubo [21]. In brief, powdered materials were soaked for seven, 14, and 21 days in simulated body fluid (SBF) with an ion concentration nearly equal to that in human blood plasma, prepared by dissolving NaCl, NaHCO_3_, KCl, MgCl_2_·6H_2_O, CaCl_2_, Na_2_HPO_4_, and Na_2_SO_4_ in ultra-pure water buffered at pH 7.4 with (4-(2-hydroxyethyl)-1-piperazineethanesulfonic acid) (HEPES) (Sigma-Aldrich, Milan, Italy).

In order to maintain the SBF solution temperature fixed at 37 °C, the samples were placed in polystyrene bottles in a water bath. As the ratio between the total surface area of the SBF-exposed material and its volume affects the reaction of hydroxyapatite nucleation, a constant ratio was maintained. Moreover, the SBF soaking solution was exchanged every two days to avoid depletion of the ionic species in SBF due to the formation of biominerals. After each soaking period, the samples were removed from the SBF and dried in a desiccator. FT-IR analysis was performed after a further 24 h, in order to evaluate the ability to form an apatite layer on the surface.

### 2.5. Antiradical Capacity

The investigation of the radical scavenging capacity of the synthesized materials took advantage of two in vitro chemical tests, which use two different probes: DPPH (2,2-diphenyl-1-picrylhydrazyl) radical and ABTS (2,2′-azinobis-(3-ethylbenzothiazolin-6-sulfonic acid)) radical cation. In both methods, powders of hybrid materials (0.5, 1.0, and 2.0 mg) were directly put in contact with the reaction mixture. In the first case, the samples were added to a methanolic solution of DPPH (9.4 × 10^−5^ M; 1.0 mL final volume), which was stirred and allowed to react for 30 min at 25 °C. After that the absorbance at 515 nm was measured in reference to a blank, using a Perkin-Elmer Victor^3^ multi-label reader. The results were expressed in terms of the percentage decrease of the initial DPPH radical absorption by the test samples [22].

In the second test, an ABTS radical cation solution was firstly generated by reacting ABTS (7.0 mM) and potassium persulfate (2.45 mM) and allowing the mixture to stand in the dark at room temperature for 12–16 h. Then, the ABTS radical cation solution was diluted in phosphate buffered saline (PBS) (pH 7.4) in order to reach an absorbance of 0.70 at 734 nm. After 6 min of direct contact between powdered materials and the reaction mixture, the absorption at 734 nm was measured by a Perkin-Elmer Victor^3^ multi-label reader in reference to a blank. The results were expressed in terms of the percentage decrease of the initial ABTS•^+^ absorption by the test samples [23].

Obtained results were expressed as mean ± standard deviation (SD) values.

### 2.6. Antibacterial Activity

Antibacterial properties of synthesized hybrid materials were evaluated against the Gram-negative *Escherichia coli* (ATCC 25922) and Gram-positive *Enterococcus faecalis* (ATCC 9212) bacteria. To this purpose, the bacterial culture was diluted in distilled water to produce a bacterial cell suspension of 10 × 10^5^ colony-forming unit (CFU)/mL. *E. coli* was inoculated in TBX Medium (Tryptone Bile X-Gluc) (Liofilchem, Roseto degli Abruzzi (TE), Italy), while *E. faecalis* was inoculated in Slanetz Bartley Agar Base (Liofilchem, Roseto degli Abruzzi (TE), Italy) in the presence of 100 mg of the hybrid powders. Afterward, the bacteria were incubated with the materials for 24 h at 44 °C and 48 h at 36 °C, respectively. The microbial growth was evaluated by observing the diameter of the inhibition halo (ID). The obtained values were expressed as mean ± standard deviation (SD) values of the measurements carried out in triplicate [24].

## 3. Results and Discussion

### 3.1. Spectroscopic Characterization of the Hybrid Materials

FT-IR spectroscopy proved to be a useful tool to get information about the structural interaction between the silica matrix and different amounts of caffeic acid within the synthetised materials. To this aim, the FT-IR spectra of silica gel (SiO_2_) and pure caffeic acid (CAF) were compared to those recorded for the hybrid materials (SiO_2_/caffeic acid at different wt.%) (Figure 2 and Figure 3). The spectral assignments of the phenol compound were based on previous literature data [25,26]. The intense bands in the FT-IR spectrum of pure caffeic acid (Figure 2a) between 4000 and 2600 cm^−1^ were assigned to the OH stretching vibrations, which partially overlaied the weak CH stretching modes of the benzene moiety and acyclic chain. The very strong band at 1645 cm^−1^ was assigned to C=O stretching vibrations, whereas those at 1625, 1530, and 1450 cm^−1^ were assigned to both aromatic and olefinic C–C stretching modes. In-plane bending modes of olefinic and aromatic C–H bonds were detected at 1280 and 1217 cm^−1^, whereas the frequencies lower than 1120 cm^−1^ referred to in-plane and out-of-plane C–C–C bending modes of the aromatic ring, with the exception of signals at 815 and 648 cm^−1^, attributable to the in-plane bending modes of the carbonyl group.

On the other hand, Figure 2c shows the FT-IR spectrum of pure SiO_2_, with the typical signals already described elsewhere [27,28]. Briefly, the broad, intense band at 3440 cm^−1^ was due to O–H stretching modes of water, whereas the bending mode was detected at 1640 cm^−1^. Hydrogen-bonded water molecules and hydrogen-bonded OH groups attached to the Si atoms were expected to occur. Symmetric and asymmetric Si–O stretching vibrations were identified at 1200 and 1080 cm^−1^, while Si–O–Si bending mode frequencies were at 800 cm^−1^ and 460 cm^−1^. Moreover, according to the literature, peaks at 1382, 960, and 570 cm^−1^ were due to residual nitrate anions resulting from HNO_3_ used as a catalyst in the synthesis procedure, as well as Si–OH bond vibrations and four-membered siloxane cycles in the silica network [18,29,30].

FT-IR spectra of the synthesized materials suggested the formation of a new network, in which inorganic and organic counterparts were strongly connected to each other. The bands observed for hybrids were almost superimposable to those of pure SiO_2_, in terms of vibration frequency and intensity. The only exception was constituted by the peaks at 1735 and 1725 cm^−1^ (Figure 2b), which appeared more intense as the caffeic acid wt.% increased (Figure 3). These signals, attributable at least in principle to an ester C=O stretching vibration, could be the proof of the formation of H-bonds with the SiO_2_ inorganic matrix.

UV–Vis spectroscopy seemed to confirm the hypothesis of a strong connection between caffeic acid and the silica matrix. As for the FT-IR spectra, in this case, the comparison of material spectra with pure inorganic and pure organic constituents dissolved in EtOH was also useful. In Figure 4a, the UV spectrum of the phenol compound was reported. Briefly, it is mainly characterized by four absorption bands at 324, 296, 242, and 216 nm. The n→π* and π→π* electronic transitions referring to the C=O group were responsible for the absorption at 296 and 216 nm, respectively, whereas the bands at 242 and 324 were attributable to the π→π* transitions of the aromatic moiety [25]. At the considered caffeic acid concentration (3.5 × 10^−5^ mol∙L^−1^) the band at the lowest wavelength was the most intense. It was previously reported that the shifts of maximum absorbance, intensity variation, and isosbestic points are due to the self-association of caffeic acid through hydrogen bonding of carboxylic groups, which lead to the formation of a dimeric structure [31].

The UV–Vis spectra of all the synthesized materials were recorded after UAE of the powders in a hydroalcoholic solution. In all cases, a hypochromic shift of the characteristic bands of caffeic acid was observed (Figure 4b). To minimize the effects of silica absorption, the SiO_2_ absorbance values were subtracted from those of each hybrid (Figure 4b, gray box). The differences observed in the spectra could be ascribed to some modifications which occurred on the phenol structure after merging inside the inorganic matrix.

### 3.2. Bioactivity

In order to evaluate the bioactivity of synthesized hybrid materials, all samples were treated according to Kokubo’s in vitro apatite forming-ability test [21]. FT-IR spectra of each material, acquired before exposure to SBF, were compared to those recorded after seven, 14, and 21 days of exposure (Figure 5). Si–OH groups of silica-based sol–gel materials were reported to induce hydroxyapatite nucleation after soaking in SBF, being able to attract the Ca^2+^ ions present in the fluid. As a consequence, a positive surface charge is enhanced. The blending of calcium cations with the phosphate anions leads to the formation of amorphous phosphate, which spontaneously transforms into hydroxyapatite [Ca_10_(PO_4_)_6_(OH)_2_] [32].

The hypothesis of hydroxyapatite precipitate formation was corroborated by data recorded. In fact, it should be noted that the band at 570 cm^−1^ split in two new peaks at lower and higher frequencies. This could be attibutable to the stretching of the hydroxyapatite –OH groups and the vibrations of PO_4_^3−^ groups caused by the formation of the hydroxyapatite precipitate [33]. Moreover, a slight up-shift of the Si–OH band (from 960 to 970 cm^−1^) suggested the interaction of the hydroxyapatite layer with the –OH groups of the silica matrix.

### 3.3. Radical Scavenging Capacity

DPPH and ABTS tests allowed us to evaluate in vitro the ability of the synthesized materials to act as chain breakers, i.e., inactivators of radical species following the transfer of a hydrogen atom (HAT) or of a single electron (SET).

Data obtained showed that the two radical probes were differently neutralized by the materials. Overall, the antioxidant capacity detected by the ABTS assay was considerably higher than that measured by the DPPH assay for all samples.

As depicted in Figure 6, pure SiO_2_ proved to be completely ineffective on both targets. Thus, the activity of hybrids could be completely ascribed to the presence of the phenol compound embedded in the silica matrix, and it is worth noting that an increase in the caffeic acid wt.% led to an increment in the antiradical capacity. Moreover, the radical scavenging capacity of hybrids was also directly correlated to the materials’ tested dose (ranging from 0.5 to 2 mg), put in direct contact with the reaction mixture. In the ABTS assay, this was particularly evident for SiO_2_/CAF10, whereas, for SiO_2_/CAF15 and SiO_2_/CAF20 samples, a plateau was reached at a radical scavenging capacity (RSC) of about 90%.

The scavenging capacity of caffeic acid can be ascribed to the *o*-dihydroxyl functional groups on the aromatic ring (also referred to as catecholic groups), as they are commonly regarded as an essential feature ruling the antiradical activity of other compounds with similar structural features [34]. Thus, it is reasonable to assume that, although caffeic acid underwent structural modifications during the material synthesis, they did not involve the catechol moiety.

### 3.4. Antibacterial Activity

The Gram-negative *Escherichia coli* and Gram-positive *Enterococcus faecalis* were used as representative bacteria to investigate antibacterial properties of the synthesized materials. To this purpose, the microbial growth was evaluated by observing the inhibition halo diameter after inoculation with 100 mg of the hybrid powders. In Figure 7, pictures of the bacterial cell suspension are reported, as well as the measured values of inhibition zones. Based on the obtained results, both bacteria seemed to be sensitive to the presence of the hybrids, with the Gram-positive bacteria slightly more responsible. The halo’s size was directly related to the amount of caffeic acid embedded in the materials, reaching maximum values of 76.4 ± 3.8 and 86.1 ± 4.3 mm for SiO_2_/caffeic acid 20 wt.% vs. *E. coli* and *E. faecalis*, respectively. It is worth noting that the activity augumented with the increase in the caffeic acid weight percentage in the synthetized hybrids, highlighting, according to previous findings, that SiO_2_ did not exert an antibacterial effect [35].

The antimicrobial potential of caffeic acid, and in some cases also its synergistic effects with antibiotics, was reported against a number of bacterial strains, including *E. coli* and *E. faecalis* [36,37]. However, the mechanism of action is not yet fully understood, as it could involve many sites at the cellular level. Several hypotheses were reported, including an altered permeability of cell membranes, the bond of phenolics to cell enzymes, and the disruption of membrane integrity, as they cause consequent leakage of essential intracellular constituents [38,39]. Also, the difference in antibacterial activity between Gram-positive and Gram-negative bacteria is still controversial in the literature. Indeed, some authors demonstrated that phenol compounds proved to be less effective vs. Gram-negative ones because of their cell wall being linked to an outer complex membrane, which slows the passage of chemicals [40,41]. On the contrary, other papers underlined that the susceptibility was mainly strain-specific [38,42]. It should be noted that, in analogy with previous hypotheses regarding 5-*O*-caffeoylquinic acid [15], the bacterial cell membrane impairment could also be the result of a decrease in reactive oxygen species (ROS) levels in the bacteria induced by caffeic acid, which leads to cellular malfunction and finally to death. This deduction is in line with the antioxidant behavior of the considered phenol.

## 4. Conclusions

The sol–gel route was successfully employed to synthesize new biocompatible and bioactive organic–inorganic hybrid materials. They differed in terms of the percentage of the organic constituent, which was represented by caffeic acid, a C_6_-C_3_ phenol produced in several fruits and vegetables. The choice of this compound relied on the knowledge that, in the pure form, it is able to exert several beneficial properties for human health. FT-IR and UV–Vis spectroscopic tools allowed us to assert that a new network was born, in which inorganic and organic counterparts were strongly connected to each other. The chemical characterization evidenced the occurrence of some structural modifications, which nevertheless ensured a strong radical scavenging capacity of samples with 15 and 20 wt.% of caffeic acid incorporated. Antimicrobial activity evaluation vs. *E. coli* and *E. faecalis* showed that both bacteria were sensitive to the presence of the hybrids, with the Gram-positive bacteria slightly more responsible.

## Figures and Tables

**Figure 1 materials-13-00394-f001:**
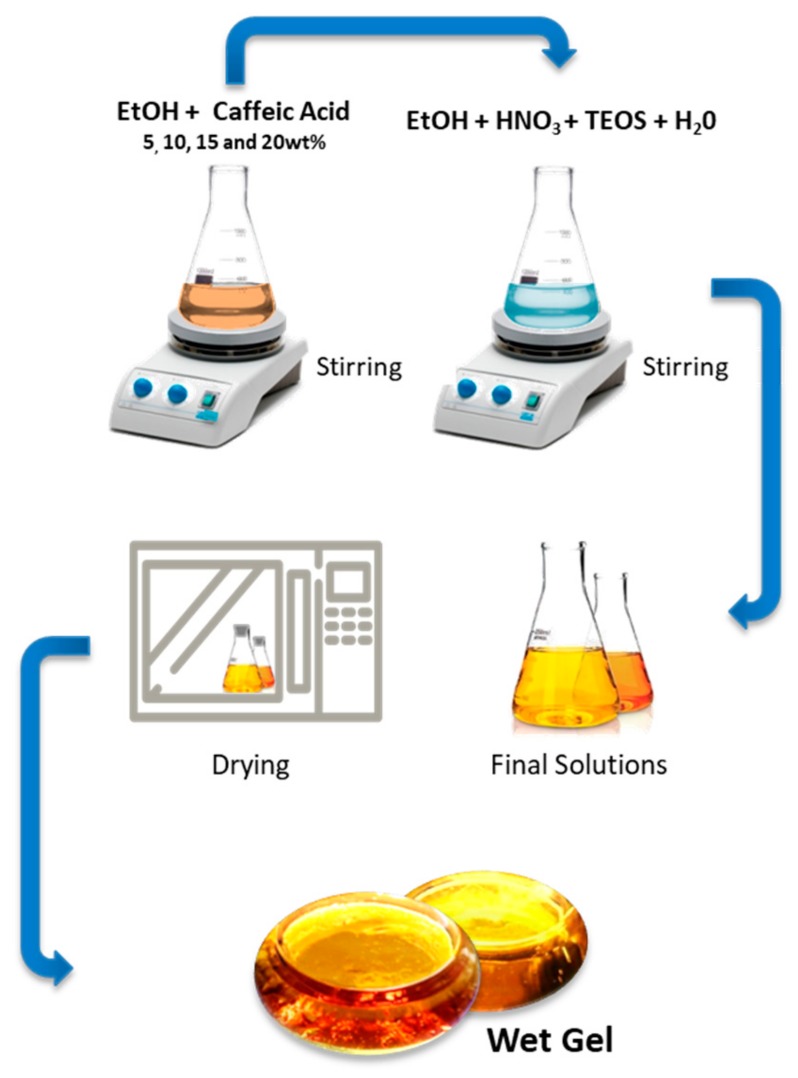
Sol–gel procedure used to obtain hybrid materials under study.

**Figure 2 materials-13-00394-f002:**
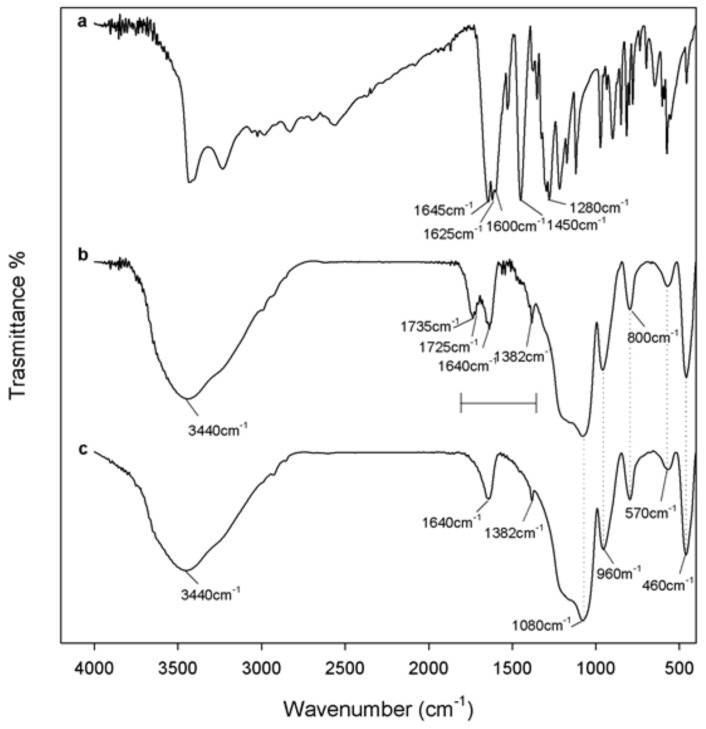
Fourier-Transform Infrared (FT-IR) spectra of (**a**) pure caffeic acid, (**b**) SiO_2_/caffeic acid 20 wt.% hybrid material, and (**c**) pure SiO_2_.

**Figure 3 materials-13-00394-f003:**
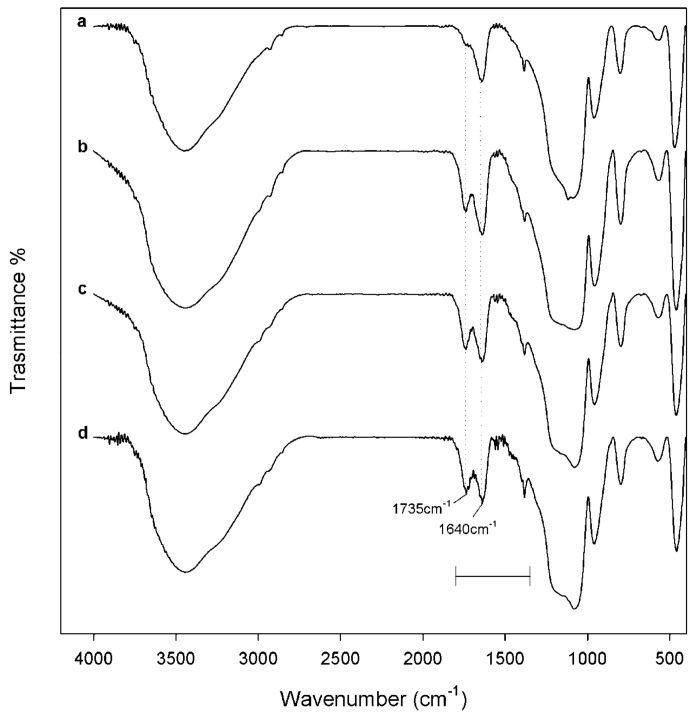
FT-IR spectra of synthesized materials: (**a**) SiO_2_/caffeic acid 5 wt.%; (**b**) SiO_2_/caffeic acid 10 wt.%; (**c**) SiO_2_/caffeic acid 15 wt.%; (**d**) SiO_2_/caffeic acid 20 wt.%. Spectra were vertically shifted to help in the comparison.

**Figure 4 materials-13-00394-f004:**
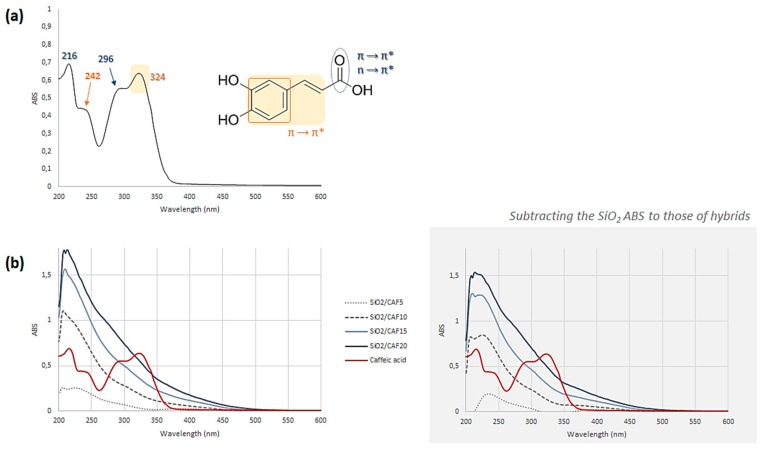
Ultraviolet-visible (UV–Vis) spectra of (**a**) pure caffeic acid dissolved in EtOH at a molar concentration of 3.5 × 10^−5^ mol∙L^−1^, and (**b**) hybrid materials differing in the amount of caffeic acid (CAF5 = caffeic acid 5 wt.%; CAF10 = caffeic acid 10 wt.%; CAF15 = caffeic acid 15 wt.%; CAF20 = caffeic acid 20 wt.%). In the gray box, UV–Vis spectra are reported after subtracting the SiO_2_ absorbance (ABS) values from those of each hybrid. Red lines refer to pure caffeic acid.

**Figure 5 materials-13-00394-f005:**
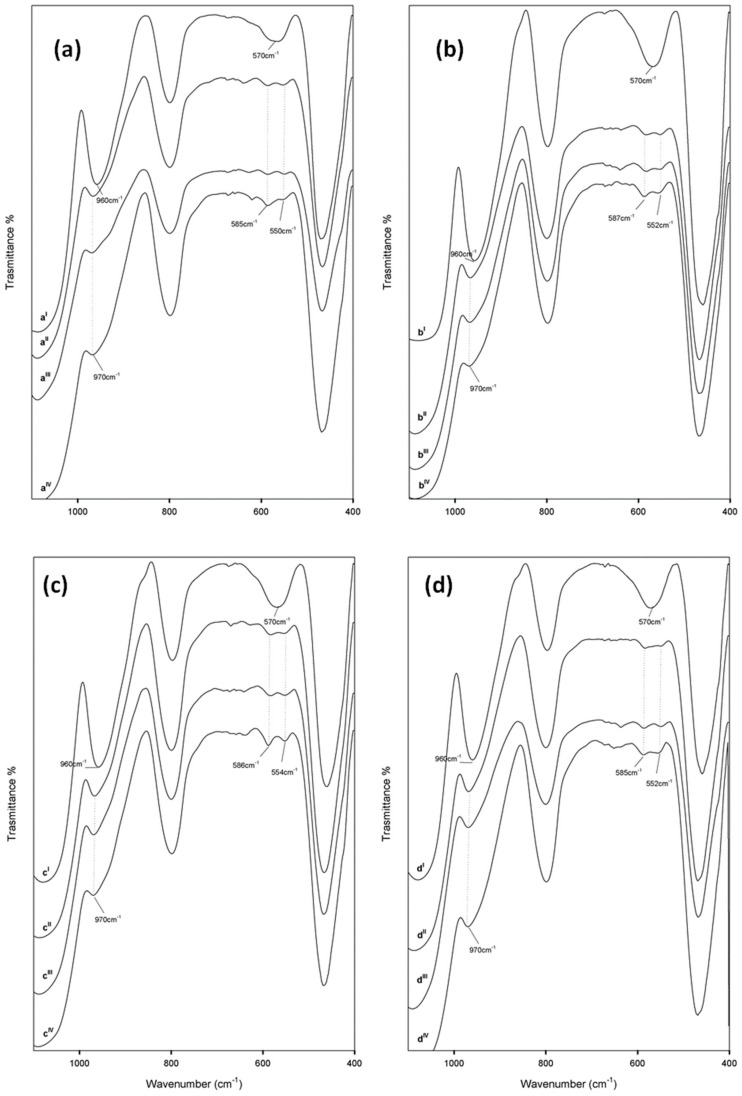
FT-IR spectra of (**a**) SiO_2_/caffeic acid 5 wt.%, (**b**) SiO_2_/caffeic acid 10 wt.%, (**c**) SiO_2_/caffeic acid 15 wt.%, and (**d**) SiO_2_/caffeic acid 20 wt.%. For each sample, spectra acquired before exposure to simulated body fluid (SBF) (I) were compared to those recorded after seven, 14, and 21 days of exposure to SBF (II–IV). Spectra were vertically shifted to help in the comparison.

**Figure 6 materials-13-00394-f006:**
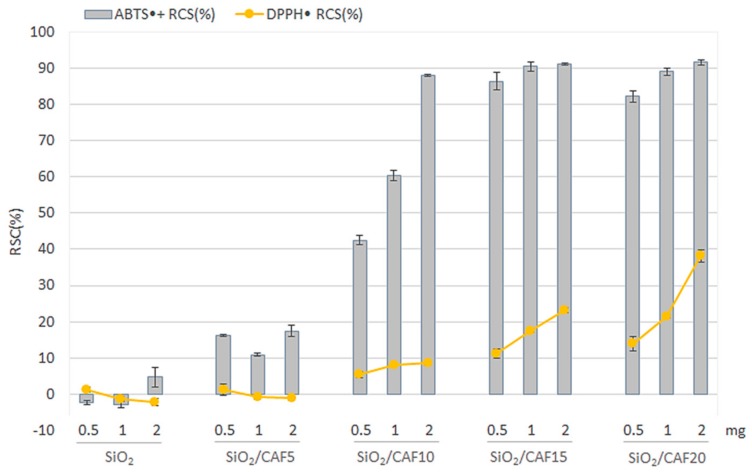
Radical scavenging capacity (RSC, %) toward 2,2-diphenyl-1-picrylhydrazyl (DPPH) radical and 2,2′-azinobis-(3-ethylbenzothiazolin-6-sulfonic acid) radical cation (ABTS^•+^) of SiO_2_ and hybrid materials differing in terms of the amount of caffeic acid (CAF5 = caffeic acid 5 wt.%; CAF10 = caffeic acid 10 wt.%; CAF15 = caffeic acid 15 wt.%; CAF20 = caffeic acid 20 wt.%). Values, reported as percentage vs. a blank, are the means ± SD of measurements carried out on three samples (*n* = 3) analyzed three times.

**Figure 7 materials-13-00394-f007:**
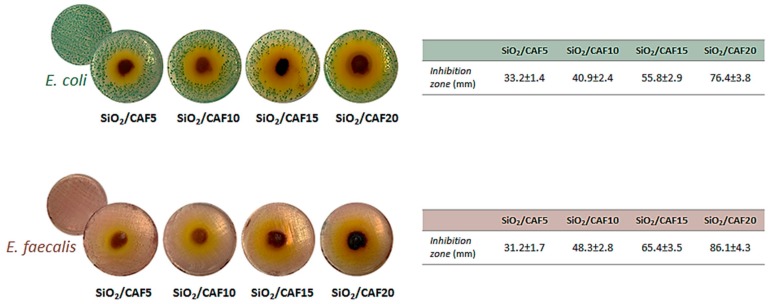
Inhibition halo (ID) of *Escherichia coli* and *Enterococcus faecalis* inoculated in the presence of synthesized materials differing in terms of the amount of caffeic acid (CAF5 = caffeic acid 5 wt.%; CAF10 = caffeic acid 10 wt.%; CAF15 = caffeic acid 15 wt.%; CAF20 = caffeic acid 20 wt.%) and inhibition zone (diameters, mm). Values are the means ± SD of measurements carried out on samples analyzed three times.

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
