# Peer review of "New SiO2/Caffeic Acid Hybrid Materials: Synthesis, Spectroscopic Characterization, and Bioactivity"

_materials, 2020, doi:10.3390/ma13020394_

Round 1

Reviewer 1 Report

This manuscript describes the preparation of new composite materials consisting of silica and caffeic acid via the sol-gel technique, followed by the characterisation of the materials via FTIR spectroscopy and UV/Vis spectroscopy and testing of the materials' radical-scavenging capacity and antibacterial activity.

I can't find any fault in the science of the manuscript. There are a number of typographical errors or stylistic errors which I list below:

line 20, "occurred" -> "occurring" line 57, "an useful" -> "a useful" line 157, "streching" -> "stretching" line 185, "isobestic" -> "isosbestic" line 190, "differing for" -> "differing in" line 196, "substracted to" -> "subtracted from" line 248, "the the", remove one "the" line 256, "wchich" -> "which" line 266 "the the" -> "that the" 

Author Response

Dear Editor,

First of all, thank you very much for giving us the possibility to revise and ameliorate our paper.

Please, find enclosed the revised version of the manuscript, as well as our response to the Reviewer's comments.

As required, we have addressed the points raised by the Reviewers and have completed the revision process. Accordingly, we are resubmitting a new version to go back to the Reviewers for re-evaluation.

Once again thanks a lot for your kind attention and assistance.

We do hope the revised manuscript will definitively meet the criteria for publication in Materials Journal.

Best regards,

Michelina Catauro,

Detailed Responses to Reviewers

Reviewer 1

This manuscript describes the preparation of new composite materials consisting of silica and caffeic acid via the sol-gel technique, followed by the characterization of the materials via FTIR spectroscopy and UV/Vis spectroscopy and testing of the materials' radical-scavenging capacity and antibacterial activity.

I can't find any fault in the science of the manuscript. There are a number of typographical errors or stylistic errors which I list below:

line 20, "occurred" -> "occurring" line 57, "an useful" -> "a useful" line 157, "streching" -> "stretching" line 185, "isobestic" -> "isosbestic" line 190, "differing for" -> "differing in" line 196, "substracted to" -> "subtracted from" line 248, "the the", remove one "the" line 256, "wchich" -> "which" line 266 "the the" -> "that the" 

The authors thank the reviewer. All typographical errors or stylistic errors have been correct according to the Reviewer’s suggestions.

Reviewer 2 Report

The authors made the research entitled "New SiO2/caffeic acid hybrid materials: synthesis, spectroscopic characterization and bioactivity". This research has certain innovation, so the reviewer suggests that after minor revision, it could be published on "Materials".

For the sol-gel route in the Introduction, several references were recommended:

Zhou, et al. Fabrication of 3D TiO2 micromesh on Silicon surface and its effects on platelet adhesion. Materials Letters, 2014, 132: 149-152.

As we known, caffeic acid has effective function on antibacterial, while what’s the role of the SiO2 in this new materials? As reported (Li, et al. The effects of Cu-doped TiO2 thin films on hyperplasia, inflammation and bacteria infection. Applied Sciences-Basel, 2015, 5(4):1016-1032.), TiO2 had similar property with SiO2. Why the authors choose SiO2, but not TiO2?

Author Response

Dear Editor,

First of all, thank you very much for giving us the possibility to revise and ameliorate our paper.

Please, find enclosed the revised version of the manuscript, as well as our response to the Reviewer's comments.

As required, we have addressed the points raised by the Reviewers and have completed the revision process. Accordingly, we are resubmitting a new version to go back to the Reviewers for re-evaluation.

Once again thanks a lot for your kind attention and assistance.

We do hope the revised manuscript will definitively meet the criteria for publication in Materials Journal.

Best regards,

Michelina Catauro,

Reviewer 2

The authors made the research entitled "New SiO2/caffeic acid hybrid materials: synthesis, spectroscopic characterization and bioactivity". This research has certain innovation, so the reviewer suggests that after minor revision, it could be published on "Materials".

For the sol-gel route in the Introduction, several references were recommended:

Zhou, et al. Fabrication of 3D TiO2 micromesh on Silicon surface and its effects on platelet adhesion. Materials Letters, 2014, 132: 149-152.

According to Reviewer's suggestions, references have been added in the Introduction section

As we known, caffeic acid has effective function on antibacterial, while what’s the role of the SiO2 in this new material?

As an increase in caffeic acid amount was relative to an augmentation in antibacterial effects, SiO2 seemed to not have a predominant role. This is in line with previous findings in which it was advocated the silicon dioxide inability to exert antibacterial efficacy. This observation, as well as the relative reference, has been added (written in red) in the text (please see paragraph 3.4).

As reported (Li, et al. The effects of Cu-doped TiO2 thin films on hyperplasia, inflammation and bacteria infection. Applied Sciences-Basel, 2015, 5(4):1016-1032.), TiO2 had similar property with SiO2. Why the authors choose SiO2, but not TiO2?

Taking into account the mechanical properties and the low production costs for silicon dioxide, the sol gel approach was firstly applied using this metal oxide. Furthermore, the previous use of SiO2 for the embedment of other phenol and polyphenol compounds (such as quercetin and chlorogenic acid), encourages us to investigate the antiradical and anti-bacterial features of hybrids in which, preserving the metal oxide component, the phenol one was represented by the simple C6C3 compound, caffeic acid.

The authors cannot deny that the preliminary biological responses of the synthetized hybrids provide an incentive for them to synthetize new materials in which the organic portion of the caffeic acid will be preserved and TiO2 will be considered as metal oxide part.